# An Investigation into the Potential of a Penicillium Commune Strain to Eliminate Aromatic Compounds

**Maria Gerginova** [1], **Katya Stoyanova** [1], **Nadejda Peneva** [1], **Ivayla Dincheva** [2] and **Zlatka Alexieva** [1,*]

1 Department of General Microbiology, Institute of Microbiology, Bulgarian Academy of Sciences, 1113 Sofia, Bulgaria; mariagg@microbio.bas.bg (M.G.); katya_litova@abv.bg (K.S.); peneva_nad@yahoo.com (N.P.)

2 Plant Genetic Research Group, AgroBioInstitute, Agricultural Academy, 1164 Sofia, Bulgaria; ivadincheva@yahoo.com

* Correspondence: zlatkama@yahoo.com; Tel.: +359-888-565-523

**Abstract:** The quantity of industrially polluted waters is increasing everywhere, of which a significant part is occupied by a number of mono- and poly-aromatic compounds. Toxins enter the soil, sewage, and clean water by mixing with or seeping into them from industrial wastewater. By using 18S RNA and ITS sequences, the *Penicillium commune* AL5 strain that was isolated from Antarctic soil was identified. This study is dedicated to exploring its capacity to metabolize hazardous aromatic compounds. The strain showed very good potential in the degradation of hydroxylated monophenols and possessed exceptional abilities in terms of resorcinol degradation. The strain's ability to metabolize 0.3 g/L of *p*-cresol at 10 °C is notable. The strain is also capable of metabolizing LMW PAHs (naphthalene, anthracene, and phenanthrene) and eliminating all three tested compounds under 23 °C, respectively, 77.5%, 93.8%, and 75.1%. At 10 °C, the process slowed down, but the degradation of naphthalene continued to be over 50%. The quantity of PAH and a few significant intermediary metabolites were determined using GC–MS analysis. Sequencing of the enzymes phenol hydroxylase and catechol 1,2-dioxygenase revealed a close association with the genes and proteins in some fungal strains that can degrade the aromatic compounds examined thus far.

**Keywords:** biodegradation; fungi; oxiganases; PAHs; GC–MS; DNA sequencing

## 1. Introduction

One of the serious problems of our time is the increase in chemical compounds dangerous to human and animal health in waste water. This happens by starting with industrial wastewater and then passing through the mixing or leaching of these toxins into the soil and clean water intended for domestic use.

A number of aromatic compounds, both mono-derivatives of phenol and polyaromatic compounds, such as naphthalene, anthracene, phenanthrene, and their higher molecular derivatives, occupy a significant share of these pollutants. Many efforts have been made to characterize microorganisms of different taxonomic affiliations possessing mechanisms to degrade phenol, catechol, cresols, chloro- and nitro-phenols, and other toxic monophenols [1–6]. Studies devoted to the removal of PAHs from sludges formed in sewage treatment plants and sewage under the influence of aerobic microorganisms are few. Some studies have shown that aeration accelerates the removal of PAHs [7,8].

An essential problem in modern biotechnology aimed at processes related to environmental cleaning is overcoming low temperatures in some polluted areas. The capacity of some microorganisms to develop and function at low temperatures is crucial to overcoming these challenges. Studies on psychrophilic and psychrotrophic microorganisms, which can degrade and utilize a variety of environmental contaminants with a diverse chemical composition, are relatively rare [9–12]. In this context, since no additional energy inputs are required for heating and maintaining relatively high temperatures for ex situ degradation,



the economic advantages of using biotechnologies are also significant. The dynamics of the ambient temperature, which occur locally and at treatment plants in a large part of the world, especially in the winter, have an impact on in situ degradation as well.

The majority of such studies have been carried out using bacteria. A number of significant results have been reported in the degradation of toxic aromatic environmental pollutants, including PAHs. A number of basidiomycetes, white rot fungi, and other filamentous fungi have also been shown to be capable of degrading PAHs and are generally more efficient than bacteria. There are significantly fewer studies devoted to the degradation potential of ascomycete fungi. Under different climatic and nutritional conditions, microbial communities with different biodiversity are formed that can mediate bioremediation processes, and fungi among them show promising potential. Fungi are among the most common microbes in the world. They stand out for their exceptional adaptability to a variety of environmental and climatic factors as well as a variety of feeding substrates. In addition to being extremely difficult to digest (such as lignin and cellulose), the chemicals and compounds used as food substrates also include a variety of highly poisonous molecules that are well-known environmental hazards. Many researchers have been inspired to concentrate on the identification and characterization of previously known and recently discovered fungal strains as a result of these skills [13–15].

Publications involving studies with different species of molds that can metabolize aromatic compounds as a carbon source occupy a significant place in the scientific literature. Numerous developments in the field of biodegradation and assimilation of toxic pollutants have been devoted to fungi, such as *Phanerochaete chrisosporium* [16,17], *Aspergillus awamori* [18], and *Fusarium* [19]. However, the use of fungi to remove aromatic contaminants is still a current issue. New strains of fungi from the genera *Fusarium* and *Alternaria* are emerging, revealing significant potential for PAH degradation. [20,21]. The majority of studies on the removal of mono- and poly-aromatic compounds by fungi concern one of the mechanisms for their removal, which is connected to the presence of a lignolytic enzyme system or extracellular peroxidases (lignin peroxidase, manganese peroxidase, and laccase) [22,23]. Although not many, there are articles that report on the uptake of aromatic compounds used as the only carbon source in fungi cultivation [24–28].

Fungi are, for the most part, aerobic microorganisms. The importance of oxygenase and dioxygenase enzymes in the aerobic catabolism of aromatic chemicals in living organisms has long been recognized [29]. With the development of molecular technologies, the potential of microbes to degrade aromatics was recognized through the identification of genes encoding proteins with hydroxylase and dioxygenase activity. Comparative analyses of homologous DNA and protein sequences might help identify potential evolutionary relationships or similar degradation mechanisms [30–32].

During a comparative analysis of the microbial diversity in wastewater and sewage sludge, it was found that representatives of the genus *Penicillium* (50%) are most often registered. This fact reveals the tolerance and potential of such strains for industrial wastewater treatment [33].

The potential of members of the *Penicillium* genus (Ascomycota) to create secondary metabolites such as antibiotics has attracted a lot of research [34,35]. *Penicillium* strains capable of growing in the presence of significant amounts of heavy metals such as Pb (II), Cu (II), and Cd (II) were isolated [36]. Much less development has been devoted to their qualities as biodegradants. Notable results have been reported demonstrating the degradation of polyethylene by *Penicillium simplicissimum* strain YK [37].

There are reports of strains from the genus *Penicillium* that actively take part in the biotransformation and degradation of aromatic compounds in the scientific literature [38,39]. Although relatively few, there are scientific papers describing the degradation of polyaromatic compounds such as anthracene, phenanthrene, pyrene, and fluorene by strains of *Penicillium* [22,40–42].

The study of the microbial diversity in Antarctica has increased over the past years. There was a significant amount of taxonomic variety discovered, including several bacterial,

yeast, and fungal species. Investigations are made into the physiological and metabolic traits of Antarctic microorganisms. Some of them have been discovered to have interesting biotechnological characteristics in addition to their capacity to survive in the extreme conditions typical of this continent. Among these microbial strains is the subject of the present study, *P. commune* AL5.

The purpose of this study is to investigate the ability of *Penicillium commune* strain AL5 to metabolize hazardous aromatic compounds at low and moderate temperatures, including phenol, catechol, resorcinol, cresol, and LMW PAHs (naphthalene, anthracene, and phenanthrene). The temperature regimes chosen (23 °C and 10 °C) were based on our desire to compare the degradation activity of the strain in mesophilic and psychrophilic conditions. In our degradation studies with the strain, we have found that at these temperature combinations, we observe good culture growth and performance and can obtain reliable comparisons for the strain degradation potential. We monitored the dynamics of their degradation as the only source of carbon. Some significant intermediate metabolites that reveal the type of PAH degradation have been found through the use of GC–MS studies. The sequencing of the putative phenol hydroxylase and catechol 1,2-dioxygenase genes in the studied strain aimed to conclusively confirm its biodegradation potential and allow comparative analysis to show the presence of phylogenetic similarity with other similar genes in fungi.

## 2. Materials and Methods

Strain *Penicillium commune* AL5 is one of 109 filamentous fungi isolated from soil samples collected at the Bulgarian Antarctic base "St. Kliment Ohridski" on Livingston Island (South Shetland Islands, Maritime Antarctica). The samples were taken from the 3-cm-deep upper soil layer. The isolates were taxonomically identified using macromorphological and micromorphological characteristics using selective media. The samples were kept and transported at 2 °C to the laboratory for further analysis. The optimal growth temperature of each selected isolate was determined by cultivating the strain at different temperatures. The studied strain shows good growth at temperatures from 10 °C to 25 °C and can be defined as psychrotrophic [43,44].

### 2.1. Media and Culture Conditions

The following nutrient media were used for its cultivation in the research process: Universal Beer Agar (Fluka, Buchs, Switzerland), Rich nutrient medium Yeast Extract Peptone Dextrose (Fluka Analytical, Sigma Aldrich, St. Louis, MO, USA)—components for 1 L: yeast extract (Difco, Beirut, Lebanon) 10 g, bactopeptone (Difco) 20 g, glucose 10 g. In the analysis of the biodegradation abilities of the strain, a minimal nutrient medium (Czapek Dox) containing the following elements for 1 L was used: 2 g $NaNO_3$; 1 g $KH_2PO_4$; 0.5 g KCl; 0.5 g $MgSO_4 \cdot 7H_2O$; 0.01 g $FeSO_4 \cdot 7H_2O$. The solid nutrient media used have the same composition as the liquid ones, with an added amount of agar (15 g per 1 L).

When analyzing the ability of the strain to grow in a medium containing the tested aromatic compounds (phenol, catechol, resorcinol, hydroquinone, *p*-cresol, naphthalene, anthracene, and phenanthrene), glucose in the medium was replaced by 0.3 g/L of each of them, individually.

The monoaromatic hydrocarbons were supplied by Merck-Schuchardt, Germany. The polyaromatic hydrocarbons naphthalene (99%), anthracene (99%), and phenanthrene (>97%) were purchased from Sigma Aldrich, St. Louis, MO, USA. The salts ($NaNO_3$, $KH_2PO_4$, KCl, $MgSO_4 \cdot 7H_2O$, and $FeSO_4 \cdot 7H_2O$) for the preparation of Czapeck Dox mineral medium were provided by Sigma Aldrich, MO, USA.

The strain was initially cultured in minimal medium (Czapek Dox) containing 1% glucose to obtain some initial biomass. To avoid the transfer of medium components, the obtained cell mass was washed with 0.9% NaCl. That cell suspension was used for inoculation in Czapek Dox medium supplied with each of the tested compounds as the sole carbon source. To ensure equal amounts of biomass, the initial $OD_{540}$ values

(Jenway 6306 spectrophotometer, Cole-Parmer Instument Co. Europe, St Neots, UK) in the experiments were adjusted to 0.235–0.31. Cultures were mixed in 200-mL flasks with an initial working volume of 20 mL on a rotary shaker (180 rpm) at 10 °C or 23 °C. The duration of the cultivation process varied depending on the experiment's goals: it lasted until the middle of the exponential phase to obtain vegetative seed material and record enzyme activity, and it lasted until the stationary phase to determine the potential for the compound's complete degradation. The centrifuged biomass is kept in storage at −20 °C as needed.

### 2.2. Analytical Methods

#### 2.2.1. Determination of the Concentration of Phenol and Phenolic Derivatives

Quantification of phenol and catechol was performed spectrophotometrically with 3,4-dimethyl aminoantipyrine at $\lambda = 540$ nm (Jenway 6305, Cole-Parmer Instument Co. Europe, St Neots, UK) [45]. The concentrations of resorcinol, hydroquinone, and *p*-cresol were determined by Folin's method. Absorbance at 620 nm was read spectrophotometrically (Jenway 6305, UK) [46].

#### 2.2.2. Extraction of PAHs and GC–MS Analyses

The extraction of high-molecular-weight polycyclic aromatic hydrocarbons (PAHs) was performed from 2-mL samples taken during the exponential growth phase. The solutions were extracted threefold with an equal volume of DCM (2 mL) for 10 min, and then the organic phases were collected and evaporated to dryness in a laboratory concentrator (RVC 2–25 CD plus, Martin Christ, Osterode am Harz, Germany) to approximately 1 mL before the samples. Prior to analysis, samples were derivatized by the following procedure: 50.0 µL pyridine and 50.0 µL N,O-Bis (trimethylsilyl)trifluoroacetamide (BSTFA) were added to a sample. The solution was heated for 1 h/75 °C/300 rpm on a thermo shaker. Finally, 1.0 µL was injected into the GC–MS system. The analyses were carried out on a 7890A gas chromatograph (Agilent Technologies Inc., Santa Clara, CA, USA) interfaced with a 5975C mass selective detector (MSD). The compounds were separated using a HP-5 ms silica fused capillary column (30 m length × 0.32 mm i.d. × 0.25 µm film thickness). The oven's temperature schedule was 50 °C for 0 s, 25 °C/s to 195 °C for 1.5 s, 8 °C/s to 265 °C for 0 s, and 20 °C/s to 315 °C for 1.25 s. The split ratio was 5:1, and the injection volume was 1.2 µL. The quadrupole, injector, and ionization source had respective temperatures of 230 °C, 150 °C, and 250 °C. The full-scan mode was used to operate the MSD. At 70 eV, the electron impact mode was used to collect all mass spectra.

By comparing their retention indices and MS fragmentation patterns with those from the National Institute of Standards and Technology (NIST'08), compounds were identified. The estimated RIs were determined using a mixture of a homologous series of aliphatic hydrocarbons from C8 to C40 under the same conditions described above. The external standard method was used for quantification of the PAHs in the samples. To produce a range of concentrations, the reference substances were precisely weighed and dissolved in dichloromethane. By mapping the peak regions against various concentrations ranging from 1.0 to 10.0 g/mL, standard calibration curves were created [47].

#### 2.2.3. Determination of the Enzyme Activities

Preparation of molds cell extracts was fraught in potassium phosphate buffer, pH 7.5, which was added to the received pelleted cell biomass. This is followed by careful and intensive grinding with the help of a porcelain pestle. The resulting mixture was centrifuged at 10,000 rpm for 20 min at 4 °C. The obtained supernatant (clarified cell lysate) was used to determine intracellular enzyme activities.

Phenol hydroxylase (EC 1.14.13.7) enzyme activity was measured spectrophotometrically ($\lambda = 340$ nm) by recording a substrate-dependent decrease in the sorption of NADPH [48].

The method provided by Varga and Neujahr, 1970 [49] was used to establish catechol1,2-dioxygenase (EC 1.13.11.1) activity using the cumulative amount of cis, cis-muconic acid detected at 260 nm.

The amount of total protein in the sample was determined by the Bradford method [50]. The enzyme activities were determined as units per 1 mg protein (U/mg protein).

The experiments for degradation and enzyme activities were conducted in triplicate. The data in the figures reflect the average values.

### 2.2.4. Isolation of DNA from Molds and PCR Sequencing Analyses

The strain *P. commune* AL5 was cultured in a liquid rich nutrient medium (YEPD) at 23 °C on a shaker at 400 rpm for 60 h. Genomic DNA from the strain was isolated by the method of Lio et al. (2000) for preparation of fungal DNA for PCR [51].

GE Health Care puReTaq Ready-To-Go PCR Beads were used in all experiments. The volume of the reaction mixture was 25 µL. The concentration of the primers in the final volume was 10 pmol. The DNA template concentration ranged from 25 ng to 50 ng.

PCR conditions: initial step, 95 °C, 5 min; 35 cycles amplification, 95 °C, 30 s; corresponding annealing temperature, 30 s (values for each primer are shown in Tables 1 and S1); 72 °C, 45 s; extension step, 72 °C, 7 min.

**Table 1.** DNA oligonucleotide primers for PCR analysis.

| Primers | Sequence (5′ → 3′) | Annealing T °C | Source |
|---|---|---|---|
| PFf | AGGGATGTATTTATTAGATAAAAAATCAA | 58 °C | [52] |
| PFr | CGCAGTAGTTAGTCTTCAGTAAATC | | |
| ITS 1 | TCCGTAGGTGAACCTGCGG | 60 °C | [53] |
| ITS 4 | TCCTCCGCTTATTGATATGC | | |

The primers used for taxonomic identification of the studied strain are reflected in Table 1.

The resulting PCR products were sent to Macrogen Europe for sequencing. The resulting individual sequences of PCR products were formatted in a form suitable for comparison by the basic local alignment search tool (BLAST) analysis with the NCBI database [54].

## 3. Results

### 3.1. Molecular Identification

In the course of this study, the strain was identified by 18S RNA and ITS sequences as representative of *Penicillium commune* and designated as AL5. DNA oligonucleotide primers and conditions for PCR analysis are shown in Table 1. The obtained sequencings are registered in the NCBI database: ITS (with a length of 503 bp) Accession: OR085315.1, GI: 2513883438, and 18S (with a length of 710 bp) Accession: OR084958.1, GI: 2513883044.

### 3.2. Degradation of Monophenols and Enzyme Analyses

Table 2 shows the results of the degradation of five monophenolic substrates (phenol, catechol, resorcinol, hydroquinone, and *p*-cresol) during cultivation of strain *P. commune* AL5 at 23 °C and 10 °C in the dark. The nutritional medium was supplemented with each of the tested substances as the only source of carbon and energy. Each of them had an initial concentration of 0.3 g/L. It can be seen that the strain grows at very different rates when different carbon sources are included in the medium (Figures S1 and S2). With the exception of *p*-cresol (up to 1.5 g/L), four of them have been entirely degraded, which should be highlighted. At the same time, a significant difference was observed in the time required for their removal from the medium. Catechol and resorcinol are totally degraded within just one day when the strain is cultivated at 23 °C. Lowering the temperature to 10 °C increases the time for complete degradation by 1.5 to 3 fold.

**Table 2.** Degradation of monophenols (0.3 g/L) as the only carbon source in the medium by *Penicillium commune* strain AL5 at temperatures of 23 °C and 10 °C.

| Substrate | 23 °C | | 10 °C | |
|---|---|---|---|---|
| | % Degradation | Time, h | % Degradation | Time, h |
| phenol | 100 | 188 | 100 | 480 |
| resorcinol | 100 | 26 | 100 | 72 |
| catechol | 100 | 24 | 100 | 36 |
| hydroquinon | 100 | 100 | 100 | 120 |
| *p*-cresol | 50 | 144 | 100 | 144 |

The time at which the studied samples were taken corresponds to the time for the established final degree of degradation of the various compounds (Table 2).

It is noteworthy that the strain used in this investigation was capable of metabolizing a methylated phenol derivative at 10 °C. At a temperature of 10 °C, this mold was able to totally remove the 0.3 g/L of *p*-cresol that was present in the growth medium. When the strain was cultivated at 23 °C, it utilized the substrate up to 0.15 g/L.

Consistent with these results, phenol hydroxylase activity with the tested monophnolic substrate was shown to be 3.46-fold higher and catechol-1,2-dioxygenase activity was 6.6-fold higher at 10 °C than it was at 23 °C (Figure 1a,b). The same occurrence was observed in the measured phenol hydroxylase activity values for hydroquinone degradation.

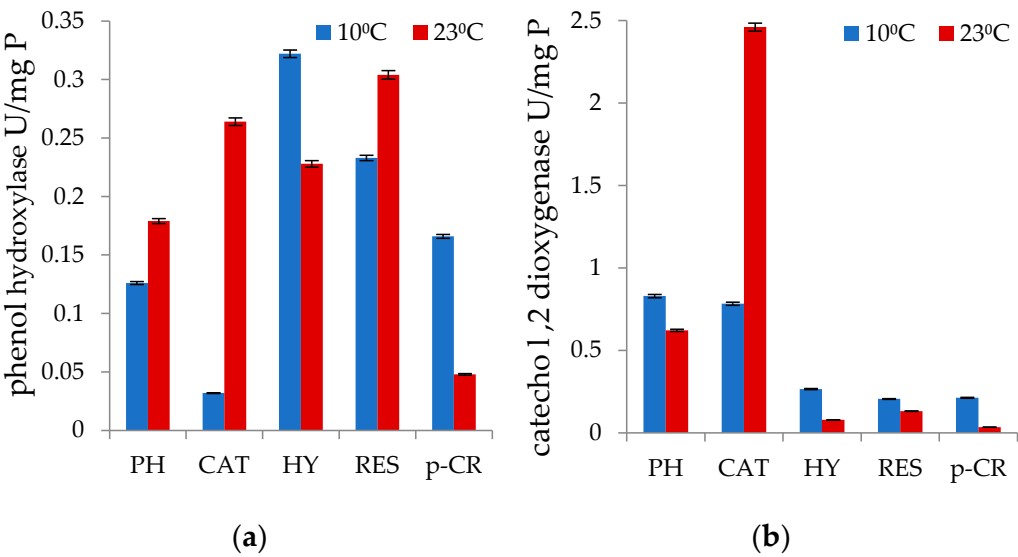

(**a**)  (**b**)

**Figure 1.** *P. commune* AL5 strain intracellular enzyme activities expressed for different carbon substrates at low and medium cultivation temperatures: (**a**) phenol hydroxylase; (**b**) catechol 1,2-dioxygenase.

*3.3. Degradation of PAHs and Catechol 1,2-Dioxygenase Activity*

*P. commune* AL5 demonstrated very good abilities related to the degradation of LMW PAHs (naphthalene, anthracene, and phenanthrene). To determine the amount of PAH in the samples, GC–MS analyses were carried out. The initial concentration of each of the examined PAH was 0.3 g/L. The strain was able to significantly eliminate all three tested compounds under mesophilic conditions (23 °C), respectively, 77.5%, 93.8%, and 75.1%. At low temperatures (10 °C), the process slowed down, but the degradation of naphthalene continued to be over 50% (Table 3).

**Table 3.** Degradation of naphthalene, anthracene, and phenanthrene as the only carbon sources in the medium by *Penicillium commune* strain AL5 at temperatures of 23 °C and 10 °C.

| Substrate | 23 °C | | 10 °C | |
|---|---|---|---|---|
| | % Degradation | Time, days | % Degradation | Time, days |
| naphthalene | 77.5 | 17 | 54 | 17 |
| anthracene | 93.8 | 17 | 8.5 | 17 |
| phenanthrene | 75.1 | 17 | 22 | 17 |

In the process of culturing and reducing the concentration of these compounds in the nutrient medium, the presence of catechol 1,2-dioxygenase activity was also checked at both temperatures. Samples for enzyme analysis were taken in the middle of the exponential growth phase of *P. commune* AL5. The data are demonstrated in Figure 2.

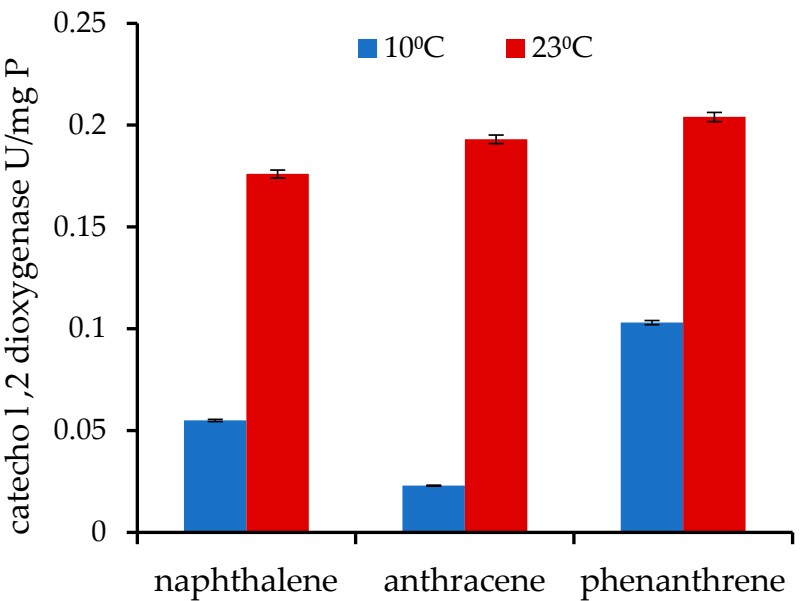

**Figure 2.** *P. commune* AL5 strain intracellular enzyme activity of catechol 1,2-dioxygenase, expressed for naphthalene, anthracene, and phenanthrene, used as sole carbon sources at low and medium cultivation temperatures.

When the individual compounds degraded at 23 °C, the values of the catechol 1,2-dioxygenase enzyme activities are not considerably varied, but at 10 °C, the values significantly fall, and the differences between them widen.

Intermediate Metabolites

The occurrence of more intermediate metabolites was found when analyzing the samples from the cultivation of *P. commune* strain AL5 in mesophilic conditions (Table 4). During the degradation of naphthalene, the presence of 1,2-dihydroxy naphthalene is identified at the beginning, and through salicyl aldehyde and salicylic acid, catechol is reached, which is degraded into substances that enter the Krebs cycle. The metabolism of phenanthrene identified 1-hydroxy-2-naphthoic acid, naphthol, *o*-phthalic acid, and protocatechuic acid. When anthracene is metabolized, 9,10-dihydroanthracene and 9,10-anthraquinone are identified.

**Table 4.** Mass spectral analysis of the metabolites found in the degradation of naphthalene, anthracene, and phenanthrene by *P. commune* strain AL5 on the fifth day of cultivation at 23 °C.

| Compound | *m/z* of Fragment Ions (Relative Abundance, %) | Structure Confirmed with a Standard |
|---|---|---|
| Naphthalene | 160(100), 131(9), 115(4), 114(5), 113(6), 88(5), 80(8), 77(9), 51(6) | 1,2-Dihydroxynaphthalene |
| | 122(100), 121(94), 104(16), 93(25), 76(26), 66(14), 65(42), 63(12), 39(41), 38(12) | Salicylaldehyde |
| | 268(8), 267(34), 135(10), 91(8), 75(8), 74(9), 73(100), 45(25), 44(4), 43(10) | Salicylic acid, TMS derivative |
| | 254(19), 239(8), 166(4), 151(6), 135(7), 75(8),74(10), 73(100), 45(14) | Catechol, TMS derivative |
| Anthracene | 324(3), 251(7), 236(6), 235(3), 178(14), 75(4), 74(7), 73(100), 45(12), 43(3) | 9,10-Dihydroanthracene, TMS derivative |
| | 208(100), 181(14),180(98), 152(78), 151(33), 150(15), 76(44), 75(17), 50(19) | 9,10-Anthraquinone |
| Phenanthrene | 188(40), 170(100), 142(42), 115(31), 114(66), 88(14), 71(17), 63(15), 57(10), 45(7) | 1-Hydroxy-2-naphtoic acid |
| | 216(100), 201(98), 185(60), 141(22), 127(21), 115(36), (73(48), 45(17) | 1-Naphthol, TMS derivative |
| | 295(16), 221(6), 149 (9), 148(16), 147(100), 141(7), 140(8), 75(6), 73(63), 45(17) | Phthalic acid, TMS derivative |
| | 371(13), 370(38), 355(22), 311(11), 281(8), 194(13), 193(9), 74(9), 73(100), 45(22) | Protocatechuic acid, TMS derivative |

### 3.4. Molecular Analyses of Phenol Hydroxylase and Catechol 1,2-Dioxigenase Genes

Primers suitable for PCR amplification and DNA sequencing were created based on DNA sequences cited in the NCBI database (Table S1).

The first four primer pairs to identify the phenol hydroxylase gene were designed based on the sequence from *P. expansum*, Acc. N XM 016744622.1 [55]. The fifth pair of primers was made using the nucleotide sequence of the *P. camemberti* complete genome, Acc. N CRL31046.1 [56].

The first pair of primers for the analysis of catechol 1,2-dioxygenase gene-specific sequences were designed based on the sequence for a gene encoding catechol 1,2-dioxygenase in *P. chrysogenum* (Acc. N XM002558549.1). The second pair of primers was designed again by using the sequenced complete genome of *P. camemberti* [56].

The DNA fragments obtained as a result of the PCR analysis were sequenced by Macrogen Europe. The PCR products for each of the two studied genes were assembled by applying the CAP3 computer program [57]. The length of the partial DNA sequence for the phenol hydroxylase gene is 3101 bp, including 5 introns, and the protein coding region (CDS) is 2184 bp. The putative catechol 1,2-dioxygenase gene sequence, including two introns, totals 891 bases. The CDS is formed by 766 bp. Analysis of the obtained sequences, after processing using BLAST and comparison with the corresponding data in NCBI, showed high homology with putative genes encoding proteins with phenol hydroxylase and catechol 1,2-dioxygenase activities in *Penicillium camemberti* (Acc. N CRL31046.1). The two sequences obtained are registered in NCBI: the phenol hydroxylase gene—Acc. N OR101821; the catechol 1,2-dioxygenase gene—Acc. N OR101820.

## 4. Discussion

The isolation and study of new and already known microbial objects in terms of their ability to remove harmful substances from the environment by metabolizing them into compounds common to living organisms are important paths to solving the problems related to a clean natural environment.

The exploitation and exploration of new territories, which are characterized by low temperatures and nutrient-poor soil, lead to the discovery of new strains with opportunities for growth and development in such harsh conditions. Numerous bacteria have been isolated and investigated, but yeasts and molds are still relatively uncommon. Many of the new microbial strains demonstrate great potential for the degradation of harmful substances, among which various monophenols and polyaromatic compounds are present [58].

Due to the wide distribution in nature of various toxic aromatic compounds, their microbial utilization and transformation into harmless components of the environment continue to be an important aspect of the research of many scientific groups around the world [59,60].

One of the most active strains, a representative of *Penicillium*, in terms of phenol degradation is able to utilize in mesophilic conditions 80% of 2 to 4 mM phenol (0.2–0.37 g/L) within 72 h. Another strain, representative of *P. chrysogenum*, degrades 0.3 g/L of phenol as the sole source of carbon and energy within 100 h. The same strain was shown to successfully degrade 0.3 g/L of catechol, resorcinol, and hydroquinone [39]. The ability of *P. commune* strains to degrade and utilize aromatic compounds has not yet been reported, despite a considerable amount of data on the degradation of aromatic compounds by various representatives of the *Penicillium* genus having already been provided in the scientific literature [61].

Based on the data available in the literature, it can be claimed that the mold strain *P. commune* AL5 emerged as an excellent biodegradant of hydroxylated monophenols, possessing outstanding capabilities in terms of resorcinol degradation. The strain's ability to adapt to low temperatures and the maintenance of its degrading functions at 10 °C are both significant characteristics. Pellet formation during cultivation in media containing toxic monoaromatic and polyaromatic compounds is an observed event in various eukaryotic microorganisms, including representatives of *Penicillium* [39,41]. *P. commune* strain AL5 also formed pellets of different sizes during cultivation in media containing each of the compounds we tested.

Much of the research devoted to the degradation of various aromatic compounds is related to the activity of lignolytic enzymes. Typically, this method is unrelated to their degradation into elements found in living organisms. Two essential enzymes of intracellular metabolism that are directly connected to the subsequent total degradation and utilization of aromatics as carbon sources are membrane-bound phenol hydroxylase and catechol 1,2-dioxygenase [7,62,63]. Both enzymes are the subject of the study that is here presented, and the results show their significant levels in the cells of *P. commune* AL5, both in mesophilic and low-temperature cultivation modes.

Observations of the functioning of enzyme systems in organisms capable of developing and functioning at lower than normal temperatures are interesting. As a rule, when the temperature decreases, there is a decrease in the levels of enzyme activity and, accordingly, in biodegradation [64,65]. A similar phenomenon was partially shown in our studies with *P. commune* strain AL5 and the first enzyme of phenol catabolism. When the strain *P. commune* AL5 was grown at a temperature of 10 °C, phenol hydroxylase showed noticeably decreased activity in the degradation of phenol, catechol, and resorcinol. However, there were two instances where the enzyme's activity was shown to increase: during the degradation of hydroquinone and *p*-cresol. As for the second enzyme in the catabolite chain, the catechol 1,2-dioxygenase enzyme activity showed higher values in the degradation of all monophenolic substrates at 10 °C (Figure 1). The results obtained demonstrate the examined strain's and its enzyme system's capacity to adapt to low temperatures without losing enzyme activity under mesophilic conditions. In-depth research has been done on the impact of low temperatures and how some microbial enzymes, such oxidases and xylanases, can maintain and improve their functionality [66,67].

The time needed for the complete degradation and assimilation of the investigated monophenolic compounds as the only carbon source is not always directly related to the measured levels of these enzyme activities. This is logical, considering the participation of other key enzymes in this process as well as the varied levels of toxicity that chemicals exert on cells before their degradation begins.

It is commonly known that PAHs are difficult to dissolve, especially those with three or more rings in their structure. Even less solubility results from lowering the temperature, making these chemicals more difficult to access and degrade [68]. Several PAH-degrading microbial strains were isolated from soils, sediments, industrial wastes, and other sources, among which the most bacteria showed different potential in degrading different PAHs [69–72]. Most fungi have been reported to require co-metabolism with another carbon source to degrade PAHs [73].

The ability of some fungi to use PAHs as a sole source of carbon and energy has also been described [24,74]. There have been relatively few reports of strains from the genus *Penicillium* that degrade PAHs. It has been demonstrated that a *P. oxalicum* strain can break down 100 mM anthracene in just 21 days [26]. *P. chrysosporium* strain may decrease 18% of the phenanthrene present in the medium to a concentration of 20 mg/L after growing for 20 days at 20 °C [21]. *Penicillium* strains were found in the leaves of plants growing close to oil refinery sites that degraded phenanthrene (79%), naphthalene (78%), anthracene (80%), and pyrene (66%) [75].

Increasingly, more attention is paid to indigenous microorganisms isolated from the cold regions of the planet. Studies on microbial strains degrading surfactants at low temperatures are important for optimizing the bioremediation of these areas as well as from the point of view of their biotechnological applications [12].

The data obtained for this study's focus, strain *P. commune* AL5, clearly demonstrate that it possesses a very good degradation capacity for degrading naphthalene, anthracene, and phenanthrene. At both low and moderate temperatures, strain *P. commune* AL5 was able to extract a significant amount of each of the three polyaromatic compounds studied when used as the only carbon source in our investigations (Table 3). These results are also in line with the observed lower effect of low temperatures on naphthalene degradation as well as the reduced solubility of anthracene compared to phenanthrene at low temperatures [64,76].

The existence and activity of the enzyme catechol 1,2-dioxygenase have been examined in a number of PAH degradation studies. The features of these compounds have been investigated in the naphthalene degradation by *Armillaria* sp. F022, the anthracene degradation by *Micobacterium fortuitum*, and the phenanthrene degradation by *Pseudomonas* sp. ZJF08 [7,77,78]. According to the information in our investigation, it is clear that this enzyme takes part in the degradation of each of the mentioned PAHs in *P. commune* AL5 cells. The values of catechol 1,2-dioxygenase activity significantly decrease at 10 °C (Figure 2), which lengthens the time required for their degradation.

The identification of several of the intermediary metabolites in the naphthalene degradation, particularly the presence of salicylic acid and catechol, convincingly proves the strain's capacity to not only remove the substance but also to use it for growth. A similar inference may be made regarding the elimination of phenanthrene, whose end metabolites identified include protocatechuic acid and *o*-phthalic acid. Only two of the initial metabolites in the chain of degradation that are present in the majority of fungi were discovered during the degradation of anthracene (Table 4). Our findings are consistent with the stated schemes for PAH breakdown in aerobic bacteria and *P. oxalicum* [26,79].

The identified oligonucleotide sequences of genes in *P. commune* AL5 (OR101821 and OR101820) with putative phenol hydroxylase and catechol 1,2-dixigenase activity, and their corresponding protein sequences were compared using BLAST against the NCBI's database of accessible amino acid sequences.

Despite the large number of studies on the degradation of aromatic compounds by eukaryotic microorganisms, information on the genes relating to degradation enzymes is not widely available in the NCBI database. Cladograms showing the phylogenetic closeness of the amino acid sequences defining the enzymes with phenol hydroxylase and catechol 1,2 dioxygenase activities in strain *P. commune* AL5 are shown in Figures 3 and 4, respectively. The partial protein sequences for both enzymes that we obtained shared the highest degree of similarity with sequences from a member of the genus *P. camemberti* (CRL31046.1 and CRL27309).

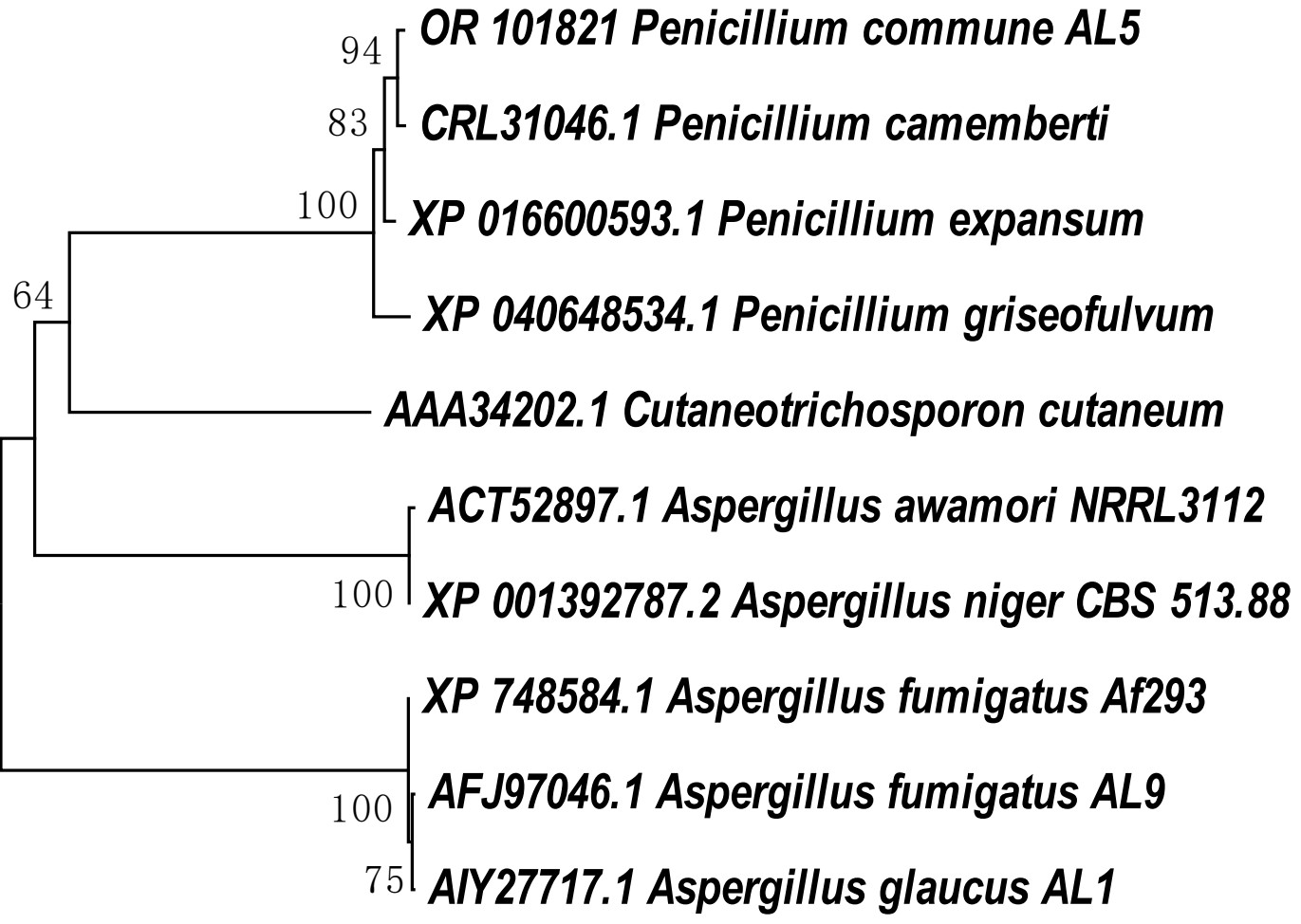

**Figure 3.** Neighbor-joining phylogenetic tree of amino acid sequences from *P. commune* AL5 phenol hydroxylase with members of the aromatic ring hydroxylase superfamily. MEGA6's ClustalW was used for sequence alignment, and MEGA6 was used to build the tree [80].

The amino acid sequences for the phenol hydroxylase from *P. commune* AL5 and *P. camemberti* have a 52% overlap, yielding a claimed 99.04% identity. When our sequence for catechol 1,2 dioxygenase was aligned with the comparable one from *P. camemberti*, there was 99.61% identity at 87% coverage of both sequences.

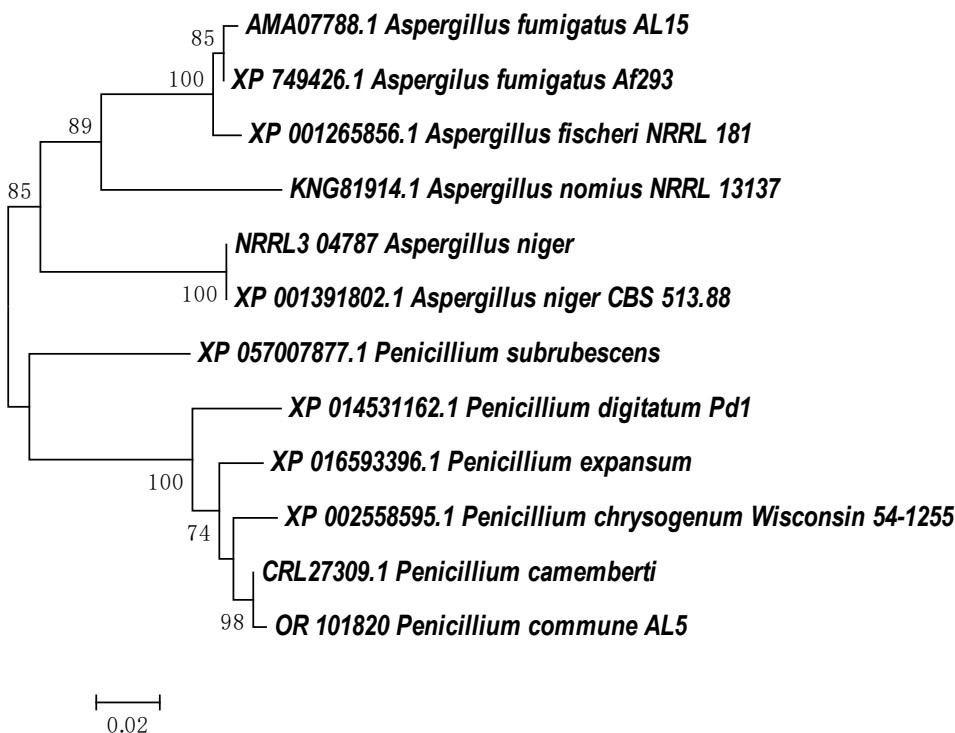

**Figure 4.** Neighbor-joining phylogenetic tree of amino acid sequences from *P. commune* AL5 catechol 1,2-dioxygenase with members of the intradiol ring cleavage dioxygenase superfamily from fungal species. MEGA6′s ClustalW was used for sequence alignment, and MEGA6 was used to build the tree [80].

This should not come as a big surprise because *P. camemberti* species are thought to have descended from the wild-type *P. commune* and share several similarities with it, including the ability to produce cyclopiazonic acid but not penicillin [81].

In the sequence of the investigated phenol hydroxylase of *P. commune* AL5 cells, two conserved motifs (GxGxxG and GD) were found in the FAD-binding region. Their presence is typical for other flavoprotein monooxygenases. They had previously been identified and researched in the yeast *Trichosporon cutaneum* ATCC46490 and the bacteria *Pseudomonas* [82,83]. The amino acid sequence for phenol hydroxylase from *P. commune* AL5 contains as well the third conserved motif (DG) for this subclass of monooxygenases identified in *Pseudomonas fluorescens* [84,85].

Four amino acid residues (Tyr-165, Tyr-199, His-223, and His-225) that are strictly specific to the enzyme's active site have been found in our partial sequence for catechol 1,2-dioxygenase. They are responsible for the coordination and activity of non-hem iron in the active center of enzymes of the dioxygenase family [31]. These features of the *P. commune* AL5 enzyme can be linked to the high catechol 1,2-dioxygenase activity seen when catechol was used as the carbon substrate.

## 5. Conclusions

In the current investigation, a highly effective fungal strain, *Penicillium commune* AL5, was identified and characterized. This species can use mono- and poly-aromatic compounds as their only carbon source at low and moderate temperatures.

This is the first report about a *P. commune* strain that utilized and completely removed phenol, catechol, resorcinol, and hydroquinone at a cultivation temperature of 23 °C and *p*-cresol at a temperature of 10 °C. It almost completely removed 0.3 g/L anthracene and over 75% of 0.3 g/L naphthalene and phenanthrene at 23 °C for 17 days and retained, albeit reduced, degradation ability at 10 °C.

Catabolic genes, such as the phenol hydroxylase and catechol 1,2-dioxygenase genes, which are members of the aromatic ring hydroxylase and the intradiol ring cleavage dioxygenase superfamilies, have been confirmed to exist by DNA and protein sequence analysis.

According to GC–MS analysis of intermediate metabolites of the studied LMW-PAHs, this strain used naphthalene via salicylic acid and catechol and phenanthrene via *o*-phthalic and protocatechuic acids, which are degraded into compounds that enter the Krebs cycle.

Representatives of the genus *Penicillium* are widespread in wastewater and sewage sludge. This shows their ability to develop in an environment containing various toxic substances and underlines their potential in industrial wastewater treatment. Our results suggest that *P. commune* strain AL5 plays an important role in the degradation of mono- and polyaromatic hydrocarbons and could be recommended for the development of biotechnologies related to the bioremediation and purification of contaminated waters and soils.

**Supplementary Materials:** The following supporting information can be downloaded at: https://www.mdpi.com/article/10.3390/pr11082402/s1.

**Author Contributions:** Conceptualization, Z.A.; methodology, M.G.; validation, Z.A.; formal analysis, N.P.; investigation, K.S., I.D. and M.G.; data curation, M.G.; writing—original draft preparation, review and editing, Z.A. All authors have read and agreed to the published version of the manuscript.

**Funding:** Project BG05M2OP001-1.002-0019: Clean technologies for sustainable environment—water, waste, energy for circular economy (Clean&Circle), for development of a Centre of Competence, is financed by the Science and Education for Smart Growth Operational programme (2014-2020), co-funded by the EU through the European Structural and Investment funds.

**Institutional Review Board Statement:** Not applicable.

**Informed Consent Statement:** Not applicable.

**Data Availability Statement:** The data presented in this study are available on request from the corresponding author.

**Conflicts of Interest:** The authors declare no conflict of interest. The funders had no role in the design of the study; in the collection, analyses, or interpretation of data; in the writing of the manuscript, or in the decision to publish the results.

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
