# Peer review of "An Investigation into the Potential of a Penicillium Commune Strain to Eliminate Aromatic Compounds"

_processes, doi:10.3390/pr11082402_

Round 1
Reviewer 1 Report
An investigation into the potential of a penicillium commune strain to eliminate aromatic compounds. Unfortunately, the following issues should be further addressed.
1.“Penicillium commune strain AL5 was isolated from soil samples from Livingstone Island, South Shetland Islands, Antarctica. It was taxonomically determined by morphological, physiological, and molecular characteristics.” Since previous studies have already established the identification of the Penicillium genus, what motivated this study to delve deeper into constructing an evolutionary tree using ITS sequences? Furthermore, ITS sequences typically provide identification at the genus level rather than the species level.
2. What factors influenced the selection of 23 and 10 degrees as reaction temperatures?
3. Since the isolation of this strain over a decade ago, what has sparked the renewed interest in studying it?
4. Table 5 placed in the supplementary document.
5. Which statistical methods were employed to analyze the data presented in the text? It is recommended to include a line representing the variance in Fig. 1 and 2.
6. A comparative analysis of the research progress made in this paper and the findings of previous studies is presented in the form of a table.
Reviewer 2 Report
The authors isolated a strain from Penicilium and they studied its degradation ability of PAHs. Their results found that AL5 can utilize and remove of PAH tested in this study under various conditions. Their results suggest the potential of using Penicilium to treat industrial wastewater.
In general, the results are interesting, but this paper needs significant improvement before publication. Please see my details comments below.
Line 49: suggest adding some sentences to show why is Fungi instead of bacteria
Line 91: would suggest adding a little bit more background info about why AL5 was isolated from Antarctic instead of the industrial wastewater.
also, explain a little bit more why you think AL5 will have PAH degradation potential? I know you mentioned that in the previous paragraph you mentioned that there are a few reports showed that Penicilium have the potential to transform/degrade PAHs.
Line 103: although your previous work has been published, still suggest adding more details to avoid your reads to go to the other paper to get more details when reading this paper.
Line 104: a lot of details are missing from this section. For example, for the testing condition, did you culture AL5 with glucose for certain to make sure you have enough AL5 and then culture AL5 with PAH? or you culture AL5 without glucose from the beginning?
for different individual PAH, did you check the initial cell density to make sure different condition has the same/similar cells? or did you standardize that when you do data analysis?
for the culture condition, did you use shaker? if so, what's the shaking speed? what's the size of the shake flasks? what's the initial working volume? what's the initial cell density? did you do media exchange?
please add more details.
Line 122: minor. leave a space between the number and the unit. Check and revise throughout this paper.
Line 134: more details are needed for the GC-MS settings, standard, etc.
Line 160: should “A5” be “AL5”?
Line 160: please explain why cells from DNA extraction has to be cultured using another media?
Line 163: please add more details about PCR conditions, besides the total volume and primer volume.
Line 165: please explain why the DNA template varies.
Line 166: can you explain why use both 18S and ITS here?
Line 178: suggest adding more info, such as the seq length and so on.
Line 184: do you have time course data? If yes, I would suggest presenting time course degradation rate. If no, can you explain why samples were collected at different time?
Line 216: same as previous section. would suggest presenting time course degradation data if you have.
Line 254: this table is confusing.
1) will you be able to identify the molecular based on the m/z?
2) The MW column info already showed in the second column (i assume the first number of the second column)
Line 264: I feel like Table 5 should be in method section instead of results section.
Line 286: the info in this section is redundant. suggest simplifying and focusing on
1) why AL5 you isolated is important to study PAH degradation?
2) what results you obtained to support the first question and how your results compared with previous studies?
3) what do your results mean to the industry?
Line 399: suggest moving figure 3 and figure 4 to results section.
Round 2
Reviewer 1 Report
The authors responded well to the reviewer's comments by the inclusion of additional experimental data.
Reviewer 2 Report
No further comments